# Clinical Management of COVID-19 in Cancer Patients with the STAT3 Inhibitor Silibinin

**DOI:** 10.3390/ph15010019

**Published:** 2021-12-24

**Authors:** Joaquim Bosch-Barrera, Ariadna Roqué, Eduard Teixidor, Maria Carmen Carmona-Garcia, Aina Arbusà, Joan Brunet, Begoña Martin-Castillo, Elisabet Cuyàs, Sara Verdura, Javier A. Menendez

**Affiliations:** 1Medical Oncology, Catalan Institute of Oncology, Dr. Josep Trueta Hospital of Girona, 17007 Girona, Spain; aroque@iconcologia.net (A.R.); eteixidor@iconcologia.net (E.T.); ccarmona@iconcologia.net (M.C.C.-G.); jbrunet@iconcologia.net (J.B.); 2Department of Medical Sciences, Medical School, University of Girona, 17003 Girona, Spain; 3Girona Biomedical Research Institute (IDIBGI), 17190 Salt, Spain; aarbusa@idibgi.org (A.A.); bmartin@iconcologia.net (B.M.-C.); ecuyas@idibgi.org (E.C.); sverdura@idibgi.org (S.V.); 4Program Against Cancer Therapeutic Resistance (ProCURE), Metabolism and Cancer Group, Catalan Institute of Oncology, 17007 Girona, Spain; 5Catalan Institute of Oncology, IDIBELL, 08908 L’Hospitalet de Llobregat, Spain; 6Unit of Clinical Research, Catalan Institute of Oncology, 17007 Girona, Spain

**Keywords:** SARS-CoV-2, COVID-19, STAT3, silibinin, Legalon, COVID-GRAM score

## Abstract

COVID-19 pathophysiology is caused by a cascade of respiratory and multiorgan failures arising, at least in part, from the SARS-CoV-2-driven dysregulation of the master transcriptional factor STAT3. Pharmacological correction of STAT3 over-stimulation, which is at the root of acute respiratory distress syndrome (ARDS) and coagulopathy/thrombosis events, should be considered for treatment of severe COVID-19. In this perspective, we first review the current body of knowledge on the role of STAT3 in the pathogenesis of severe COVID-19. We then exemplify the potential clinical value of treating COVID-19 disease with STAT3 inhibitors by presenting the outcomes of two hospitalized patients with active cancer and COVID-19 receiving oral Legalon^®^—a nutraceutical containing the naturally occurring STAT3 inhibitor silibinin. Both patients, which were recruited to the clinical trial SIL-COVID19 (EudraCT number: 2020-001794-77) had SARS-CoV-2 bilateral interstitial pneumonia and a high COVID-GRAM score, and showed systemic proinflammatory responses in terms of lymphocytopenia and hypoalbuminemia. Both patients were predicted to be at high risk of critical COVID-19 illness in terms of intensive care unit admission, invasive ventilation, or death. In addition to physician’s choice of best available therapy or supportive care, patients received 1050 mg/day Legalon^®^ for 10 days without side-effects. Silibinin-treated cancer/COVID-19+ patients required only minimal oxygen support (2–4 L/min) during the episode, exhibited a sharp decline of the STAT3-regulated C-reactive protein, and demonstrated complete resolution of the pulmonary lesions. These findings might inspire future research to advance our knowledge and improve silibinin-based clinical interventions aimed to target STAT3-driven COVID-19 pathophysiology.

## 1. STAT3 and COVID-19 Pathophysiology

Coronavirus disease 2019 (COVID-19) is a severe acute respiratory syndrome coronavirus 2 (SARS-CoV-2)-induced inflammatory disease of the airways and lungs that can lead to multiorgan damage and death [1,2]. In severe cases, COVID-19 pathophysiology involves the impairment of type I interferon (IFN-I) production accompanied by acute respiratory distress syndrome (ARDS) and extensive endotheliopathy and coagulopathy [3,4,5,6,7,8]. An imbalance in the opposing roles of transcription factors STAT1 and STAT3 seems to operate as the central triggering event at the initial site of SARS-CoV-2 infection in the respiratory epithelium [9,10,11,12,13], resulting in catastrophic inflammatory and coagulopathy/thrombosis episodes in patients at high-risk of COVID-19 (Figure 1).

During SARS-CoV-2 infection, several viral proteins antagonize the IFN-I production pathway and the downstream activation of JAK (Janus kinase)-STAT1 signaling. The inhibitory effects on IFN-1/STAT1 signaling at the site of initial infection block the inducible amplification of IFN-I-driven antiviral responses in proximal target cells, allowing the virus to spread and replicate without limitation [12,13]. Notably, whereas SARS-CoV-2 proteins inhibit IFN-I/STAT1 signaling, the capacity of SARS-CoV-2-associated molecules (single- and double-stranded RNAs and viral proteins) to stimulate the expression and secretion of proinflammatory cytokines and chemokines remains intact. These events chronically activate the cytokine release syndrome hyperimmune response (a.k.a. *cytokine storm*) resulting from the unrestricted engagement of the high viral load to pattern recognition receptors in target cells. The vicious cycle of SARS-CoV-2 infection and damage of alveolar epithelial cells triggers the activation of monocytes, macrophages, and dendritic cells, which release chemokines (e.g., IL-6) to recruit additional immune cells that further exacerbate lung inflammation [14,15,16,17,18]. The cytokine storm ultimately instigates a systemic amplification cascade via *cis* and *trans*-presentation signaling in inflammatory immune cells as well as *trans* signaling in nonimmune cell compartments such as endothelial cells, overall contributing to key pathophysiologic phenotypes, including ARDS and thromboembolic events.

In *cis* signaling, the proinflammatory cytokine interleukin-6 (IL-6) binds to membrane-bound IL-6R (mIL-6R), which is restricted largely to innate (neutrophils, macrophages, and natural killer cells) and acquired (B and T cells) immune cells, in a complex with the ubiquitously expressed gp130. In *trans* signaling, high concentrations of circulating IL-6 can bind the soluble form of IL-6R (sIL-6R), thereby forming a complex with gp130 in cell surfaces lacking mIL-6R, such as endothelial cells (Figure 2). In *trans* presentation signaling, IL-6 binding to mIL-6R on an immune cell forms a complex with gp130 on T_h_17 cells to downstream activate T cell signaling involved in ARDS (Figure 2).

Once STAT1 function is impaired and the cytokine storm begins, a concomitant and compensatory STAT3-dependent transcriptional profile becomes dominant to further inhibit the STAT1-mediated IFN-I response. Aberrant transcriptional rewiring towards STAT3 has emerged as the master effector/mediator of all the different (*cis* and *trans*-presentation) types of the signaling nodes in type 2 alveolar cells, macrophages, extracellular matrix, T-lymphocytes, and blood, consequently triggering the majority of symptoms observed in hospitalized patients with COVID-19, including proinflammatory conditions, profibrotic status, T-cell lymphopenia, and rapid coagulopathy/thrombosis [13] (Figure 2). Importantly, a positive feedback loop is established between STAT3 activation and plasminogen activator inhibitor-1 (PAI-1; SERPIN E1), a serine protease inhibitor secreted by vascular endothelial cells that regulates fibrinolysis and exacerbates the progression of systemic inflammation, especially intravascular coagulation [19,20,21,22]. The escalating activation of the STAT3-PAI-1 interactome in multiple cellular compartments drives a catastrophic cascade of life-threatening systemic events of inflammation, fibrosis, and coagulopathy/thrombosis characteristics seen in severe cases of COVID-19.

## 2. The SilCOVID-19 Trial: Testing the Clinical Value of Natural Inhibitor of STAT3 against Severe COVID-19 in Cancer Patients

If the over-stimulation of the STAT3 signaling network is a shared node of COVID-19 pathophysiology, the application of anti-STAT3 therapies to block the SARS-CoV-2 lifecycle might moderate the severity of COVID-19. Most direct STAT3 inhibitors (STAT3i) have yet to enter clinical evaluation, early-phase clinical trials have produced mixed results with STAT3-targeted cancer therapies and, despite decades of research, very few FDA-approved STAT3i are available. The unique characteristics of the flavonolignan silibinin—the major bioactive component in the silymarin extract obtained from the seeds of the milk thistle herb (*Silybum marianum*)—as a bimodal, direct STAT3i impeding the activation, dimerization, nuclear translocation, DNA-binding, and transcriptional activity of STAT3 both in vitro and in situ [23,24,25,26], might help to circumvent the considerable complexity of targeting aberrant STAT3 signaling (Figure 3).

Indeed, silibinin has three major attributes that qualifies it as a strong candidate to clinically manage SARS-CoV-2/COVID-19 severity and mortality from a multitarget perspective [12] (Figure 4, *top*). First, it might exert direct antiviral effects including prevention of virus entry via inhibition of clathrin-mediated endocytosis and inhibition of viral replication by impeding viral RNA-dependent RNA polymerase [12,27,28,29,30,31]. Second, it functions as a direct STAT3i with proven STAT3-related therapeutic activity in cancer patients with advanced systemic disease when used orally as part of bioavailable formulations accompanied by low toxicity and reversible mild side-effects [23,24,25,26] (Figure 3). Finally, it might exert protective effects against the ability of central mediators of inflammatory responses to activate critical factors in endotheliopathy and coagulopathy, such as PAI-1 itself or the IL-6 *trans*-signaling-PAI-1 axis [32].

Based on the aforementioned rationale, we designed a randomized, open-label, phase II multicentric clinical trial (SIL-COVID19; EudraCT number: 2020-001794-77) to evaluate the therapeutic efficacy of Legalon^®^—a silibinin-containing milk thistle extract—in the prevention of ARDS in cancer patients with moderate-to-severe COVID-19 undergoing systemic treatment or having completed treatment <1 year ago [12]. SIL-COVID-19 was designed in two phases: a nonrandomized safety phase followed by a two-stage randomized phase with a prospective control (Figure 4, *bottom*). Due to slow accrual, however, the study closed prematurely after two of the eighty-two planned patients were allocated to receive silibinin combined with best supportive care according to the physician’s choice.

### 2.1. Outcomes of Two Hospitalized Patients with Active Cancer and COVID-19 Orally Receiving the Silibinin-Containing Nutraceutical Legalon^®^

We present the outcomes of two hospitalized patients with active cancer and COVID-19 receiving Legalon^®^ (Figure 5). We defined critical COVID-19 illness as a composite of admission to the intensive care unit (ICU), invasive mechanical ventilation, or death using the COVID-GRAM tool—a risk score predictor of critical illness among patients hospitalized with COVID-19—was originally developed from 10 independent predictors including chest radiography abnormality, age, hemoptysis, dyspnea, unconsciousness, number of comorbidities, cancer history, neutrophil-to-lymphocyte ratio, lactate dehydrogenase, and direct bilirubin [33,34,35].

#### 2.1.1. Patient #1

A 75-year-old male, former smoker with stage IV pancreatic cancer, tested positive for COVID-19 by PCR. The patient had a previous story of hypertension, pathological fracture of L3 due to bone metastasis, and active Psoas abscess caused by *Klebsiella oxytoca* treated with piperacillin-tazobactam. The patient presented with dyspnea for three days after he tested positive for COVID-19 without experiencing other obvious symptoms. Chest radiography revealed the presence of bilateral pneumonia (Figure 6). Basal blood tests showed several alterations, namely, lymphocytes 3.9%, hemoglobin 8.7 g/dL, dimer D 524 ng/mL, fibrinogen 691 mg/dL, albumin 2.2 gr/dL, LDH 234 U/L, C-reactive protein (CRP) 8.85 mg/mL, and troponin T 32.9 ng/mL (Table 1).

The patient received Legalon^®^ 1050 mg/day for 10 days without any side-effects. The treatment was discontinued after the patient showed improvement. The patient also received a 5-day course of remdesivir (single 200 mg loading dose on day 1 given by intravenous infusion followed by once-daily maintenance of 100 mg from day 2 by intravenous infusion) according to the guidelines on the treatment and management of hospitalized patients with COVID-19 in our institution. The patient also received prophylactic anticoagulation with heparin during hospitalization and prednisone 1 mg/kg for 10 days. The patient required supplemental oxygen at a flow rate of 4 L/min during the episode.

Follow-up chest X-ray radiography 14 days after the onset of symptoms showed complete radiographic resolution of the pulmonary lesions (Figure 5). The inflammatory marker LDH returned to reference values and CRP notably declined at the end of Legalon^®^ treatment. The patient was discharged and alive at 30 days after starting the assigned study medication.

#### 2.1.2. Patient #2

A 70-year-old male, current smoker (two packs a day) with stage IV gastric cancer tested positive for COVID-19 by PCR. The patient had a previous history of hypertension, type 2 diabetes mellitus, obesity, and chronic pulmonary obstructive disease. The patient presented with dyspnea and chest radiography established the presence of bilateral pneumonia (Figure 6).

Blood tests at the inclusion showed several alterations, namely, hemoglobin 10.1 g/dL (13.5–18), dimer D 294 ng/mL (0–230), fibrinogen 654 mg/dL (150–450), albumin 2.8 g/dL (3.5–5.2), LDH 283 U/L (135–225), CRP 10.5 mg/mL (0–0.5), and troponin T 34.5 ng/mL (<14).

The patient received Legalon^®^ 1050 mg/day for 10 days without any side-effects. The treatment was discontinued after the patient showed improvement. The patient also received prophylactic anticoagulation with heparin during hospitalization. The patient solely required supplemental oxygen (4 L/min) during the episode.

Follow-up chest X-ray radiography 14 days after onset of symptoms showed radiographic suppression and improvement of the pulmonary lesions (Figure 5). The CRP sharply declined at the end of Legalon^®^ treatment. The patient was discharged and alive at 30 days after starting the assigned study medication.

## 3. Silibinin-Treated Cancer/COVID-19+ Patients: Clinical Lessons Learned and Conundrums

Patients with both cancer and COVID-19 infection often have a more severe clinical course with worse outcomes, and lethality rates of up to 25%. Indeed, the risk of adverse outcomes of SARS-CoV-2 infection is significantly higher for patients with cancer than for the general population across a broad spectrum of malignancies; the patients are often immunocompromised and older (aged ≥ 60 years) with one or more comorbidities [36,37,38,39,40,41,42,43,44,45]. Because of the higher risk of severe evolution of COVID-19, a more intensive surveillance strategy should be incorporated into the clinical management of cancer patients (especially those with active malignancy), including early evaluation of symptoms and early treatment for COVID-19. Most reports of COVID-19 among cancer patients mainly focus on its epidemiological and clinical features [36,37], and there are few reports regarding cases of patients who contracted COVID-19 with active cancer disease and additional comorbidities.

When the COVID-19 pandemic overwhelmed the healthcare systems in Spain, particularly during the first and second waves of the pandemic, the SIL-COVID19 study was initially designed to reduce the risk of critical COVID-19 illness from 30% to 15% in a onco-hematological population, which usually have limited access to ICU and invasive ventilation due to cancer prognosis. The recent COVID-19 and Cancer Consortium (CCC19) cohort study involving patients from the USA, Canada, and Spain revealed that, among patients with cancer and COVID-19, 30-day all-cause mortality was high and associated not only with general risk factors but also with active cancer (progressing versus remission, odds ratio 5.20, 95% confidence interval 2.77–9.77) [36]. According to the cohort study of cancer patients with COVID-19 in Europe, active malignancy (*p* < 0.0001) emerged also as the sole oncologic feature predictive of higher mortality rates alongside age ≥65 (*p* < 0.0001) and comorbidities (*p* = 0.002) [37]. Systemic inflammation is another validated bedside predictor of adverse outcomes in cancer patients with COVID-19; accordingly, hypoalbuminemia (escape of albumin to the interstitial space due to inflammation-increased capillary permeability) and lymphocytopenia (reduction of lymphocytes due to systemic inflammation and direct neutralization) are independently predictive of severe COVID-19 in cancer patients, as computed by the OnCovid Inflammatory Score [46,47]. In the present case reports, cancer patients #1 and #2, both with active malignancy and systemic inflammation including hypoalbuminemia and lymphocytopenia, scored 193 and 200 points, respectively, in the COVID-GRAM risk score (88.9% and 91.7% of risk of critical illness). We acknowledge, however, that there is a need for caution when using the China cohort-based COVID-GRAM tool for predicting critical illness in patients hospitalized with COVID-19 in Europe because, while useful for identifying Caucasian patients who are at low risk of critical illness and mortality, it seems to overestimate risk in the highest-risk patients [34,35].

Silibinin-treated cancer/COVID-19+ patients exhibited a slight but significant decline in LDH, an inflammatory prognostic biomarker that is increased during acute and severe lung damage and capable of predicting with high accuracy the in-hospital mortality in severe and critically ill patients with COVID-19 [48,49]. They further exhibited a sharp decline in CRP, a reliable marker of acute inflammation that is transcriptionally activated by STAT3 in response to IL-6 [50,51]. Because CRP is a useful marker of the IL-6 signaling via the IL-6R/JAK1/STAT3 signaling pathway [52,53,54], it is tempting to suggest that silibinin is reducing IL-6-induced CRP expression via suppression of STAT3 hyperactivation. LDH and CRP appear to reflect the respiratory destress consequent to the abnormal inflammation status induced by SARS-CoV-2 infection and predict respiratory failure in COVID-19 patients [55]. Accordingly, cancer/COVID-19+ patients treated with Legalon^®^ did not require closer respiratory monitoring or more aggressive supportive therapies during their episodes, and chest radiography follow-up showed complete resolution of the initially revealed bilateral pneumonia. We cannot establish a causal relationship between a silibinin-based treatment and the favorable outcome of hospitalized, SARS-CoV-2-infected cancer patients at high risk of critical COVID-19 illness. Moreover, the concomitant receipt of remdesivir and prednisone might associate with clinical improvement in Legalon^®^-treated patient #1. Although we are lacking a matched retrospective cohort of cancer patients with bilateral pneumonia as a result of COVID-19, it should be noted that a majority of cancer patients develop at least one complication from COVID-19, the most common being acute respiratory failure and ARDS [40]. Moreover, significantly higher mortality rates are observed among male cancer patients, for those aged ≥65 years, and in those with ≥2 preexisting comorbidities [40], a threefold at-risk condition occurring in the cancer/COVID-19+ patients treated with silibinin-containing Legalon^®^.

The SIL-COVID19 trial was closed before the first scheduled interim analysis because of poor recruitment that was not amenable to improvement. The trial received approval from the Spanish Agency of Medicine and Health Products (AEMPS) in June 2020 when the first wave of the COVID-19 pandemic was in remission in Spain. Moreover, the safety cohort of the study was approved solely for a single center and, because of the re-organization of the Public Health System in Spain, several county hospitals referred patients to other centers that were designed for attending COVID-19 patients after the first wave. Refusal to participate in the trial was higher than expected; unwillingness to participate was likely multifactorial but involved both the isolation of COVID-19-positive cancer patients and the exacerbated shortage of health workers that left oncology care facilities short-staffed. Finally, the introduction of vaccinations rapidly progressed among prioritized cancer patients who received vaccination with COVID19 mRNA vaccines (Pfizer-BioNTech COVID-19 Vaccine Comirnaty^®^ and Moderna Spikevax^®^ COVID-19 vaccine) from March 2021 to May 2021. Consequently, the number of oncology patients requiring hospitalization for SARS-CoV-2 infection in the trial region dropped markedly and steady enrollment of trial patients became virtually impossible. In the light of the growing evidence that cancer patients might not respond adequately to COVID-19 vaccination [56,57,58,59,60], together with the recent discovery that a STAT3-related immune and proliferation transcriptional network is activated in COVID-19 cancer patients [61], it is important to broaden knowledge of the clinico-molecular interactions between SARS-CoV-2 and cancer at the level of STAT3. Along this line, the antidepressant fluvoxamine, which has recently been shown to significantly reduce the need for hospitalization in high-risk COVID-19 patients [62], is known to inhibit STAT3 activity [63,64].

## 4. Materials and Methods

### 4.1. Subjects

The SilCOVID-19 study was registered with the EU Clinical Trials Register and is available online [65]. The primary endpoint was to reduce the proportion of patients requiring mechanical ventilation during COVID-19 disease. Secondary aims included mortality, safety, and identification of biomarkers.

### 4.2. Legalon^®^ Dosing

The SilCOVID-19 study involved the usage of a high-dose (1050 mg/day) instead of a standard-dose (450 mg/day) of Legalon^®^ for up to 14 days. A safety cohort phase was proposed to involve 10 patients being treated in a single center to allow for response and toxicity analysis before proceeding into randomized phase II.

Previous studies have used higher doses of Legalon^®^ than those indicated on the technical data sheet. A dose of 700 mg/day vs 420 mg/day vs placebo for 48 weeks (*n* = 72 patients) or 700 mg/day versus 420 mg/day versus placebo for 24 weeks (*n* = 154) in patients with hepatitis C virus (HCV) demonstrated a safety profile comparable to placebo in both cohorts [66,67]. In a study for decompensated hepatic cirrhosis due to HCV (*n* = 62), a Legalon^®^ dosage of 1050 mg/day vs 420 mg/day was compared [68]. The high-dose cohort benefited from better clinical activity and improved quality of life without serious adverse effects. Higher doses of Legalon^®^ have been administered intravenously (20 mg/kg/ day for 14 days, which would be equivalent to 1400 mg/day for a 70 kg person) in 16 patients with HCV/HIV coinfection, showing good clinical activity and favorable toxicity profile [69]. In a cancer treatment setting, silibinin doses of 2800 mg/day (oral silybin-phosphatidylcholine) have been administered for 4 weeks with neo-adjuvant intentions to breast cancer patients (*n* = 12) without toxicity issues [70]. High-doses (13 gr/day) of oral silybin-phytosome have been administered to prostate cancer patients (*n* = 6) for 14–31 days [71]. Only one patient experienced a thromboembolic event, whereas the rest of toxicities were diarrhea (*n* = 4) and transient grade 2 hyperbilirubinemia (*n* = 1) [71].

### 4.3. Ethics Statement

The hospital (Dr. Josep Trueta Hospital, Girona, Spain) ethics committee (Clinical Investigation Ethic Committee, CIEC) approved the protocol and any amendments. All procedures were in accordance with the ethical standards of the institutional research committees and with the 1964 Helsinki declaration, and its later amendments, or comparable ethical standards. Written informed consent was obtained from all individual participants included in the study.

## 5. Conclusions

Our findings from the clinical management of COVID-19 with silibinin in hospitalized cancer patients, which should be viewed as observations rather than recommendations for treatment, might inspire future silibinin-based trails in nononcological populations of COVID-19 patients. This is the case of the SIL-COVINT-21 trial (NCT04816682), which is ongoing in a nononcological population in Slovakia. This prospective open label study will explore whether silibinin can improve the evolution of COVID-19 in patients admitted to internal medicine wards using a historical cohort for propensity-match analysis. In the active-comparator arm, consecutively-admitted patients (*n* = 30) will be allocated to receive silymarin tablets (150 mg each) three times daily (3-2-2) and will be compared with consecutive patients with the same inclusion/exclusion criteria as in the active arm, hospitalized at the same department before the initiation of the study (historical controls). Findings from this trial might help clarify whether silibinin-based approaches can therapeutically integrate the mechanisms of action of IL-6-targeted monoclonal antibodies (e.g., tocilizumab, sarilumab, siltuximab) and pan-JAK1/2 inhibitors (e.g., baricitinib) to limit the cytokine storm, T-cell lymphopenia, and coagulopathy in the clinical setting of severe COVID-19.

## Figures and Tables

**Figure 1 pharmaceuticals-15-00019-f001:**
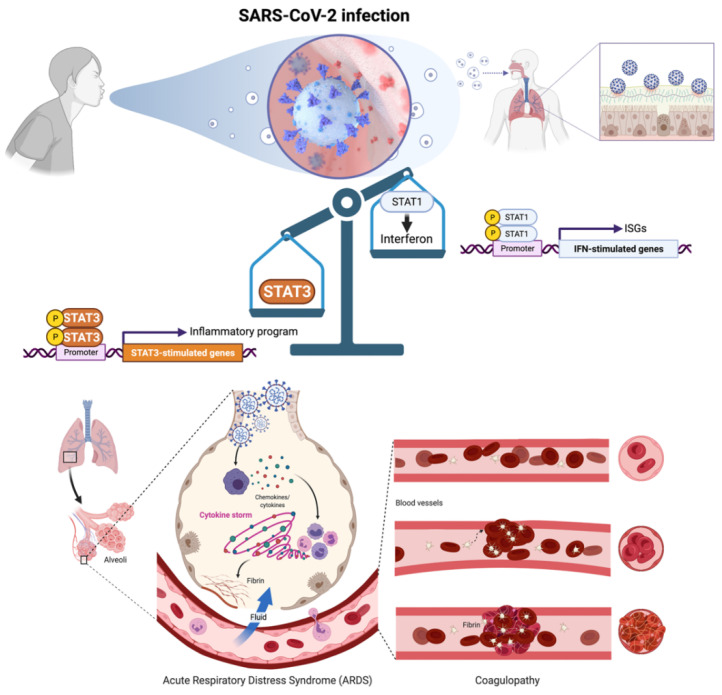
**SARS-CoV-2 infection-driven STAT1-to-STAT3 transcriptional shifting is central to COVID-19 pathophysiology.** SARS-CoV-2-mediated inhibition of STAT1/IFN-I-stimulated genes (ISGs) and the subsequent shift to a hyperactive, STAT3 dominant proinflammatory and immune response signaling may drive the key clinical features of severe COVID-19 including acute respiratory distress syndrome (ARDS) and coagulopathy/thrombosis events. Prevention of the excessive, compensatory activation of STAT3 that occurs once STAT1 is compromised and symptoms arise after SARS-CoV-2 infection, is a therapeutic avenue that might impede the combinatorial activation of both ARDS-driving inflammatory cytokines/chemokines and plasminogen activator inhibitor-1 (PAI-1)-related coagulopathy/thrombosis events in severe COVID-19 cases. Created with Biorender.com.

**Figure 2 pharmaceuticals-15-00019-f002:**
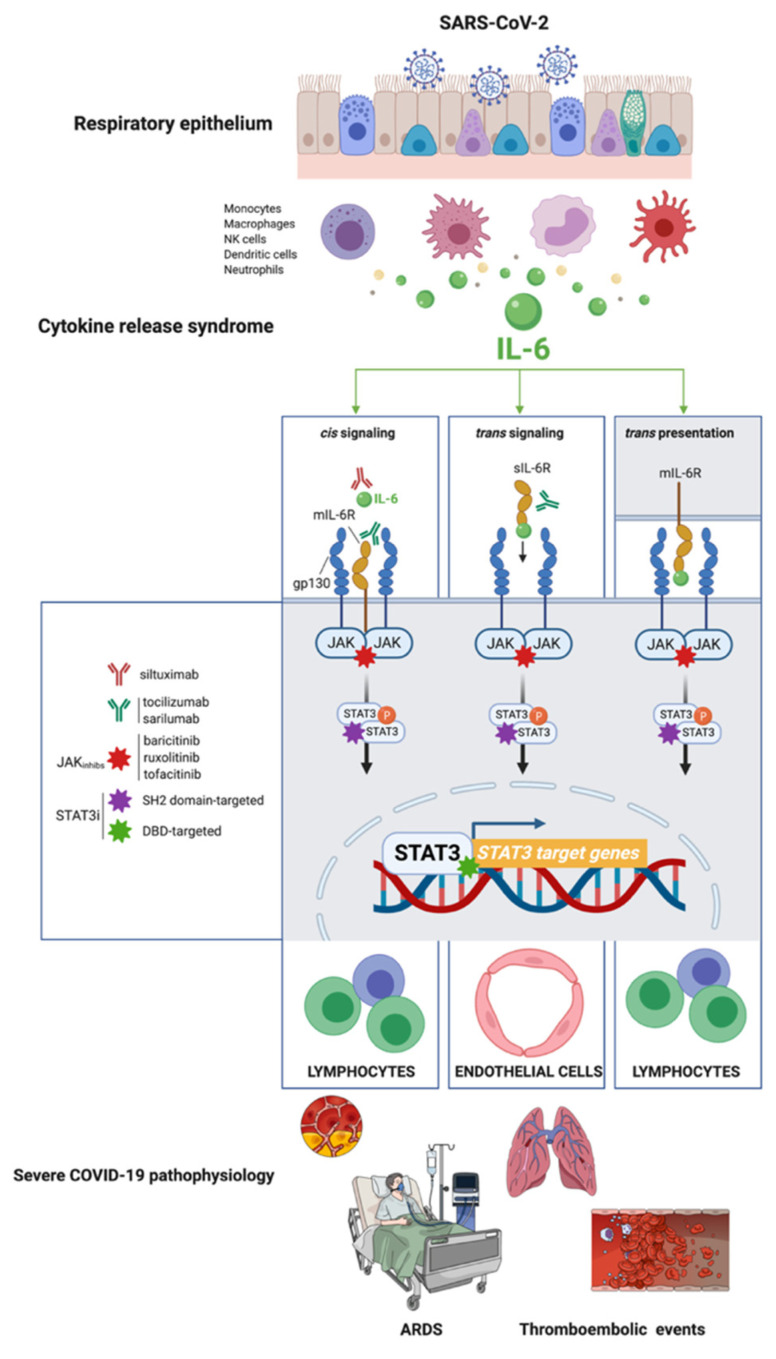
**STAT3: A master effector/mediator of the pathophysiological traits of severe COVID-19.** IL-6 inhibitors can suppress only *cis* and *trans* signaling; IL-6R inhibitors can suppress not only *cis* and *trans* signaling but also *trans* presentation. Similar to IL-6/IL-6R inhibitors, pharmacological intervention with direct STAT3 inhibitors is expected to suppress IL-6-driven *cis*, *trans*, and *trans* presentation signaling in immune and nonimmune cell compartments. Direct STAT3 inhibitors (STAT3i) could be viewed as a novel strategy to prevent the activation of self-propagating, deleterious cascades of systemic inflammation that underlie the pathophysiology of pulmonary dysfunction in ARDS, vascular permeability and leakage, thrombosis, pulmonary embolism, and coagulopathy in the late scenario of severe COVID-19. Created with Biorender.com and mindthegraph.com.

**Figure 3 pharmaceuticals-15-00019-f003:**
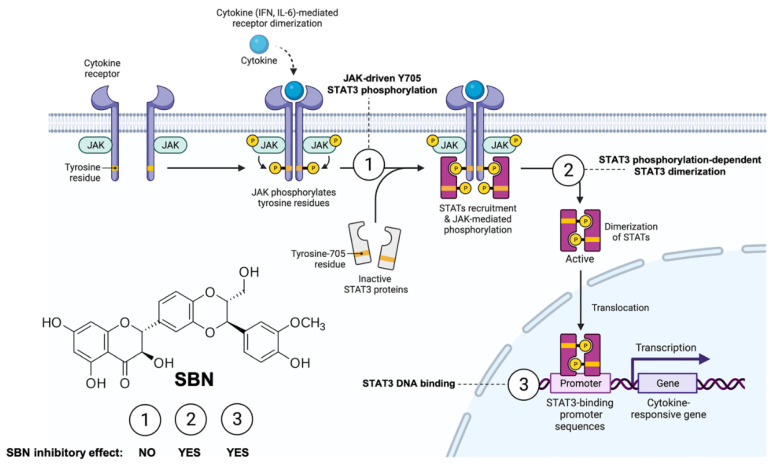
**Silibinin: A bimodal STAT3i of the JAK-STAT3 pathway.** Silibinin can reduce IL-6 inducible, constitutive, and acquired feedback activation of STAT3 at tyrosine 705 (Y705). A multifaceted combination of enzymatic assays, computational modeling, and in vitro/in situ validation has delineated the mechanism of action through which silibinin targets STAT3 [23,24,25,26]. Silibinin fails to drastically alter the activity of the STAT3 upstream kinases JAK1 and JAK2 (mechanism 1). Silibinin could directly bind to both the Src homology-2 (SH2) domain to indirectly prevent Y705 phosphorylation via a unique binding mode that partially overlap with the cavity occupied by other STAT3 inhibitors (mechanism 2) and the DNA-binding domain (DBD) of STAT3 to block the binding of activated STAT3 to its consensus DNA sequence via a unique mechanism that might involve direct interactions with DNA (mechanism 3). Accordingly, silibinin treatment suffices to reduce phosphor-activation of Y705, prevent STAT3 nuclear translocation, and suppress STAT3-directed transcriptional activity. Created with Biorender.com.

**Figure 4 pharmaceuticals-15-00019-f004:**
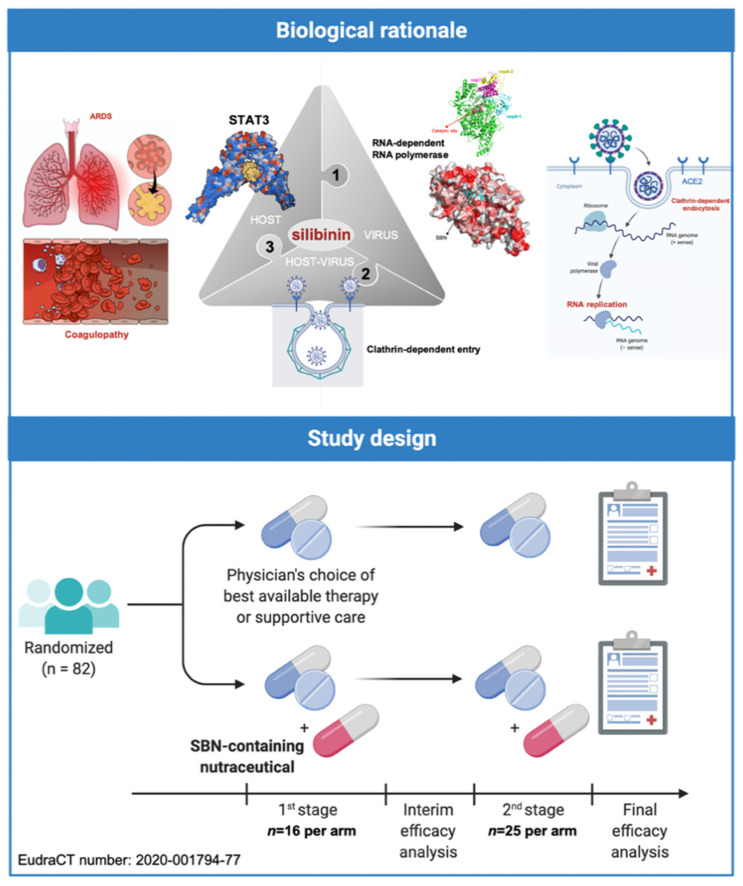
**The SilCOVID-19 trial: Biological rationale and study design.** The SilCOVID-19 study is a phase II clinical trial to evaluate the efficacy of silibinin supplementation in the prevention of the respiratory failure progression in patients with active cancer disease and infection by SARS-CoV-2. Created with Biorender.com and mindthegraph.com.

**Figure 5 pharmaceuticals-15-00019-f005:**
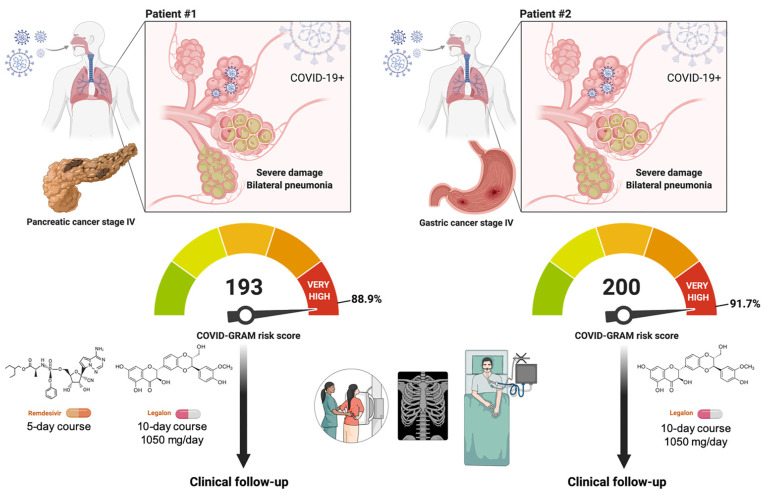
**Clinical management of COVID-19 in hospitalized cancer patients with the STAT3i silibinin: two case reports.** Both patients had SARS-CoV-2 bilateral interstitial pneumonia and a high COVID-GRAM score (i.e., 193 and 200 are the points scored in the COVID-GRAM Critical Illness Risk Score Tool by patient #1 and #2, respectively) [33,34,35]. Both patients were, therefore, predicted to be at high risk of critical COVID-19 illness in terms of intensive care unit admission, invasive ventilation, or death. In addition to physician’s choice of best available therapy (e.g., remdesivir) or supportive care (e.g., prophylactic anticoagulation with heparin), patients received 1050 mg/day Legalon^®^ for 10 days without side-effects. Created with Biorender.com.

**Figure 6 pharmaceuticals-15-00019-f006:**
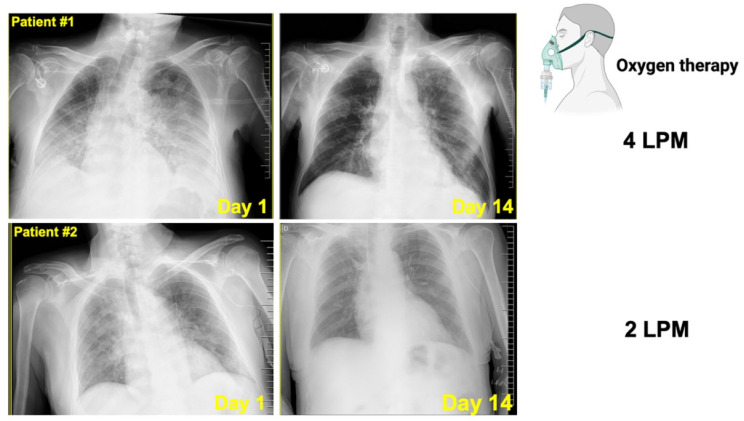
**Radiological aspects of COVID-19 pneumonia in two cancer patients treated with silibinin.***Top.* Chest X-ray series in a 75-year-old male with stage IV pancreatic cancer and COVID-19 pneumonia. *Left.* Chest X-ray depicting bilateral lung infiltrates on illness day 3. *Right.* Chest X-ray depicting complete radiographic resolution of the pulmonary lesions after 10 days treatment with 1050 mg/day Legalon^®^. *Bottom.* Chest X-ray series in a 70-year-old male with stage IV gastric cancer and COVID-19 pneumonia. *Left.* Chest X-ray depicting bilateral lung infiltrates on illness day 1. *Right.* Chest X-ray depicting complete radiographic resolution of the pulmonary lesions after 10 days treatment with 1050 mg/day Legalon^®^ (LPM: Liters per minute, L/min).

**Table 1 pharmaceuticals-15-00019-t001:** Baseline and follow-up values of serum biochemistry in silibinin-treated cancer/COVID-19 + patients.

			Patient #1			Patient #2	
	Reference	Baseline	7-Days	14-Days	Baseline	7-Days	14-Days
**LDH ***	135–225 U/L	234	245	214	283	292	232
**CRP**	0–0.5 mg/mL	8.85	7.51	4.57	10.5	7.11	1.36
**Lymphocytes**	25–40%	3.9%	10.3%	10.7%	25.3%	16.3%	21%
**Fibrinogen**	140–450 mg/dL	691	722	700	654	659	582
**Dimer D**	0–230 ng/mL	524	577	463	294	345	393
**Troponin T**	<14 ng/mL	32.9	76.9	42.1	34.5	35.4	24.8

* LDH: Lactate Dehydrogenase.

## Data Availability

The data used to support the findings of this study are included within the article. We deleted all information that might identify the patients.

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
