# Peer review of "Clinical Management of COVID-19 in Cancer Patients with the STAT3 Inhibitor Silibinin"

_pharmaceuticals, 2021, doi:10.3390/ph15010019_

Round 1

Reviewer 1 Report

In this report Bosh-Barrera and col., reported the outcome of two cancer patients from severe covid by administration of the drug silibinin.

While the article title and the abstract do very well which could be a meaningful study with a promising option to treat severe covid, the report fails to show any clear correlation between the effects of the silibinin and the outcome of the patients.

The article organization is not clear, is it a review or a report? Figures 1 and 2 center on the introductory part. Nevertheless, the information is not complete.  A detailed description of the molecular pathways and information regarding the drug silibinin could be helpful if the authors decided to elaborate a review. Figures 3 and 4 aim to describe the study design, which seems to be a broad study in progress with many patients.

The authors then describe the effects in two patients. The structure of the article also fails to fix with any of the case report journal style, since they should have provided detailed information about the patients and tested the possible correlation of the treatment with those parameters that are expected/predicted to be related with silibilin. As describe in the report it is very difficult to drag any conclusion from only two patients.

Evaluation of the possible effects of the drug over the STAT3 pathway could have been shown. Do the authors evaluate IL-6 levels in blood, or measured another possible output of the silibinin effects over the STAT3 axis?

Unfortunately, the final impression is that this description could be an anecdotic observation.

Reviewer 2 Report

It is a really common problem to hospitalize patients with both cancer and severe Covid19.

The authors have submitted their two-case experience in the treatment of STAT3 inhibitor in patients with stage IV cancers and COVID19, revealing a good outcome after 2 weeks of treatment. Unless this result should be validate with a larger experience, I have found this paper interesting to read and the analysis well conducted and argued.

Reviewer 3 Report

In this work entitled “Clinical management of COVID-19 in hospitalized cancer patients with the STAT3 inhibitor silibinin: Two case reports”, authors evaluated the therapeutic effect of silibinin, the major bioactive component in extracts obtained from the seeds of Silybum marianum, in the prevention of ARDS in patients with moderate-to-severe COVID-19. The present case report shows the outcomes of two hospitalized patients with active cancer (pancreatic cancer stage IV and gastric cancer stage IV) and COVID-19 receiving Legalon. Although the case report is interesting some concerns listed below should be addressed for its publication.

Major concerns:

- Why does the article refer to onco-hematological patients if the two patients included in the study had solid tumors instead of hematological cancer?

- Please, include a summary of the phase I clinical trial of silibinin in the discussion. Add information about the selected Legalon dose.  

- If available, a comparison with a historical cohort, age-matched, oncological patients with bilateral pneumonia upon COVID should be performed.

- Is it possible to include other measurements related to inflammation or STAT3 pathway activation using samples from the patients that received silibinin?

- In the website of the clinical trial, a 30-day secondary endpoint includes mortality rate. What is the live/dead status of the patients that received Legalon at day 30 ? This information should be included in the main text.

Minor concerns:

- COVID-GRAM score should be defined in the introduction or at the beginning of the results section.

- It would be good to incorporate the data of the blood test  (lines 172-175 and 192-195) in a table.

- Lines 211-213:  References should be cited for this statement.

- Line 266: “In the light of the growing evidence that onco-hematologic patients might not respond adequately to COVID-19 vaccination”. References should be cited for this statement.

- Fig. 2. Please, improve the legend of this figure. The legend must be limited to the description of the figure. Lines 92-101 should be incorporated in the main text.

- Fig. 3. The study design is not clear enough, what is the difference between 16 and 25 patients in each protocol arm?

- Fig. 4 y 5. The text of the figures states that patients received Legalon for 10 or 14 days, while the description of each patient in the main text states that both patients received a 10-day treatment of Legalon. Please check this information and perform corrections accordingly.

Round 2

Reviewer 1 Report

The authors have made a substantial modification in the organization of the manuscript to report their information as a Perspective. In general, the report in this version has improved considerably and it is not confusing in the style. 

Minor observations

Figure legend 2. L120: "STAT3is" is it correct?

Figure 3. Consider including captions in the figure and in the figure legends to facilitate the description and comprehension.

Considerer to mention the contribution of the effects promoted by the other drugs administrated to the patient 1 (remdesivir, prednisone) to his/her final output. 
